# BA-Net: Dense Bundle Adjustment Networks

**Chengzhou Tang**
School of Computer Science
Simon Fraser University
`chengzhou_tang@sfu.ca`

**Ping Tan**
School of Computer Science
Simon Fraser University
`pingtan@sfu.ca`

## ABSTRACT

This paper introduces a network architecture to solve the structure-from-motion (SfM) problem via feature-metric bundle adjustment (BA), which explicitly enforces multi-view geometry constraints in the form of feature-metric error. The whole pipeline is differentiable, so that the network can learn suitable features that make the BA problem more tractable. Furthermore, this work introduces a novel depth parameterization to recover dense per-pixel depth. The network first generates several basis depth maps according to the input image, and optimizes the final depth as a linear combination of these basis depth maps via feature-metric BA. The basis depth maps generator is also learned via end-to-end training. The whole system nicely combines domain knowledge (i.e. hard-coded multi-view geometry constraints) and deep learning (i.e. feature learning and basis depth maps learning) to address the challenging dense SfM problem. Experiments on large scale real data prove the success of the proposed method.

## 1 INTRODUCTION

The Structure-from-Motion (SfM) problem has been extensively studied in the past a few decades. Almost all conventional SfM algorithms (Agarwal et al., 2011; Wu et al., 2011; Schönberger & Frahm, 2016; Engel et al., 2018; Delaunoy & Pollefeys, 2014) jointly optimize scene structures and camera motion via the Bundle-Adjustment (BA) algorithm (Triggs et al., 2000; Agarwal et al., 2010), which minimizes the geometric (Agarwal et al., 2011; Wu et al., 2011; Schönberger & Frahm, 2016) or photometric (Engel et al., 2014; 2018; Delaunoy & Pollefeys, 2014) error through the Levenberg-Marquardt (LM) algorithm (Nocedal & Wright, 2006). Some recent works (Ummenhofer et al., 2017; Zhou et al., 2017; Wang et al., 2018) attempt to solve SfM using deep learning techniques, but most of them do not enforce the geometric constraints between 3D structures and camera motion in their networks. For example, in the recent work DeMoN (Ummenhofer et al., 2017), the scene depths and the camera motion are estimated by two individual sub-network branches.

This paper formulates BA as a differentiable layer, the BA-Layer, to bridge the gap between classic methods and recent deep learning based approaches. To this end, we learn a feed-forward multilayer perceptron (MLP) to predict the damping factor in the LM algorithm, which makes all involved computation differentiable. Furthermore, unlike conventional BA that minimizes geometric or photometric error, our BA-layer minimizes the distance between aligned CNN feature maps. Our novel feature-metric BA takes CNN features of multiple images as inputs and optimizes for the scene structures and camera motion. This feature-metric BA is desirable, because it has been observed by Engel et al. (2014; 2018) that the geometric BA does not exploit all image information, while the photometric BA is sensitive to moving objects, exposure or white balance changes, etc. Most importantly, our BA-Layer can back-propagate loss from scene structures and camera motion to learn appropriate features that are most suitable for structure-from-motion and bundle adjustment. In this way, our network hard-codes the multi-view geometry constraints in the BA-Layer and learns suitable feature representations from training data.

We strive to estimate a dense per-pixel depth, because dense depth is critical for many tasks such as object detection and robot navigation. A major challenge in solving dense per-pixel depth is to find a compact parameterization. Direct per-pixel depth is computational expensive, which makes the network training intractable. So we train a network to generate a set of basis depth maps for an arbitrary input image and represent the result depth map as a linear combination of these basis

depth maps. The combination coefficients will be optimized in the BA-Layer together with camera motion. This novel parameterization guarantees a smooth depth map with good consistency with object boundaries. It also reduces the number of unknowns and makes dense BA possible in networks.

Similar depth parameterization is introduced in a recent work, CodeSLAM (Bloesch et al., 2018). The major difference is that our method learns the basis depth map generator through the gradients back-propagated from the BA-Layer, while CodeSLAM learns the generator separately and uses its results for a standalone optimization component. Thus, our basis depth map generator has the chance to be better trained for the SfM problem. Furthermore, we use a different network structure to generate basis depth maps. CodeSLAM employs a variational auto-encoder (VAE), while we use a standard encoder-decoder. This design enables us to use the same backbone network for both feature learning and basis depth map learning, making joint training of the whole network possible.

To demonstrate the effectiveness of our method, we evaluate on the ScanNet (Dai et al., 2017a) and KITTI (Geiger et al., 2012) dataset. Our method outperforms DeMoN (Ummenhofer et al., 2017), LS-Net (Clark et al., 2018), as well as several conventional baselines. Due to page limit, we move the ablation studies, evaluation on DeMoN's dataset, multi-view SfM (up to 5 views), and comparison with CodeSLAM on the EuroC dataset (Burri et al., 2016) to the appendix.

## 2 RELATED WORK

**Monocular Depth Estimation Networks** Estimating depth from a monocular image is an ill-posed problem because an infinite number of possible scenes may have produced the same image. Before the raise of deep learning based methods, some works predict depth from a single image based on MRF (Saxena et al., 2005; 2009), semantic segmentation (Ladický et al., 2014), or manually designed features (Hoiem et al., 2005). Eigen et al. (2014) propose a multi-scale approach for depth prediction with two CNNs, where a coarse-scale network first predicts the scene depth at the global level and then a fine-scale network will refine the local regions. This approach was extended in Eigen & Fergus (2015) to handle semantic segmentation and surface normal estimation as well. Recently, Laina et al. (2016) propose to use ResNet (He et al., 2016) based structure to predict depth, and Xu et al. (2017) construct multi-scale CRFs for depth prediction. In comparison, we exploit monocular image depth estimation network for depth parameterization, which only produces a set of basis depth maps and the final result will be further improved through optimization.

**Structure-from-Motion Networks** Recently, some works exploit CNNs to resolve the SfM problem. Handa et al. (2016) solve the camera motion by a network from a pair of images with known depth. Zhou et al. (2017) employ two CNNs for depth and camera motion estimation respectively, where both CNNs are trained jointly by minimizing the photometric loss in an unsupervised manner. Wang et al. (2018) implement the direct method (Steinbruecker et al., 2011) as a differentiable component to compute camera motion after scene depth is estimated by the method in Zhou et al. (2017). In Ummenhofer et al. (2017), the scene depth and the camera motion are predicted from optical flow features, which help to make it generalizing better to unseen data. However, the scene depth and the camera motion are solved by two separate network branches, multi-view geometry constraints between depth and motion are not enforced. Recently, Clark et al. (2018) propose to solve nonlinear least squares in two-view SfM using a LSTM-RNN (Hochreiter et al., 2001) as the optimizer.

Our method belongs to this category. Unlike all previous works, we propose the BA-Layer to simultaneously predict the scene depth and the camera motion from CNN features, which explicitly enforces multi-view geometry constraints. The hard-coded multi-view geometry constraints enable our method to reconstruct more than two images, while most deep learning methods can only handle two images. Furthermore, we propose to minimize a feature-metric error instead of the photometric error in (Zhou et al., 2017; Wang et al., 2018; Clark et al., 2018) to enhance robustness.

## 3 BUNDLE ADJUSTMENT REVISITED

Before introducing our BA-Net architecture, we revisit the classic BA to have a better understanding about where the difficulties are and why feature-metric BA and feature learning are desirable. We only introduce the most relevant content and refer the readers to Triggs et al. (2000) and Agarwal et al. (2010) for a comprehensive introduction. Given images $\mathbb{I} = \{I_i | i = 1 \cdots N_i\}$, the geometric

BA (Triggs et al., 2000; Agarwal et al., 2010) jointly optimizes camera poses $\mathbb{T} = \{\boldsymbol{T}_i | i = 1 \cdots N_i\}$ and 3D scene point coordinates $\mathbb{P} = \{\boldsymbol{p}_j | j = 1 \cdots N_j\}$ by minimizing the re-projection error:

$$\mathcal{X} = \text{argmin} \sum_{i=1}^{N_i} \sum_{j=1}^{N_j} \|e_{i,j}^g(\mathcal{X})\|, \tag{1}$$

where the geometric distance

$$e_{i,j}^g(\mathcal{X}) = \pi(\boldsymbol{T}_i, \boldsymbol{p}_j) - \boldsymbol{q}_{i,j}$$

measures the difference between a projected scene point and its corresponding feature point. The function $\pi$ projects scene points to image space, $\boldsymbol{q}_{i,j} = [x_{i,j}, y_{i,j}, 1]$ is the normalized homogeneous pixel coordinate, and $\mathcal{X} = [\boldsymbol{T}_1, \boldsymbol{T}_2 \cdots \boldsymbol{T}_{N_i}, \boldsymbol{p}_1, \boldsymbol{p}_2 \cdots \boldsymbol{p}_{N_j}]^\top$ contains all the points' and the cameras' parameters. The general strategy to minimize Equation (1) is the Levenberg-Marquardt (LM) (Nocedal & Wright, 2006; Lourakis & Argyros, 2005) algorithm. At each iteration, the LM algorithm solves for an optimal update $\Delta\mathcal{X}^*$ to the solution by minimizing:

$$\Delta\mathcal{X}^* = \text{argmin} \|J(\mathcal{X})\Delta\mathcal{X} + E(\mathcal{X})\| + \lambda\|D(\mathcal{X})\Delta\mathcal{X}\|. \tag{2}$$

Here, $E(\mathcal{X}) = [e_{1,1}^g(\mathcal{X}), e_{1,2}^g(\mathcal{X}) \cdots e_{N_i,N_j}^g(\mathcal{X})]$, and $J(\mathcal{X})$ is the Jacobian matrix of $E(\mathcal{X})$ respect to $\mathcal{X}$, $D(\mathcal{X})$ is a non-negative diagonal matrix, typically the square root of the diagonal of the approximated Hessian $J(\mathcal{X})^\top J(\mathcal{X})$. The non-negative value $\lambda$ controls the regularization strength. The special structure of $J(\mathcal{X})^\top J(\mathcal{X})$ motivates the use of Schur-Complement (Brown, 1958).

This geometric BA with re-projection error is the golden standard for structure-from-motion in the last two decades, but with two main drawbacks:

- Only image information conforming to the respective feature types, typically image corners, blobs, or line segments, is utilized.
- Features have to be matched to each other, which often result in a lot of outliers. Outlier rejection like RANSAC is necessary, which still cannot guarantee correct result.

These two difficulties motivate the recent development of direct methods (Engel et al., 2014; 2018; Delaunoy & Pollefeys, 2014) which propose the photometric BA algorithm to eliminate feature matching and directly minimizes the photometric error (pixel intensity difference) of aligned pixels. The photometric error is defined as:

$$e_{i,j}^p(\mathcal{X}) = I_i(\pi(\boldsymbol{T}_i, d_j \cdot \boldsymbol{q}_j)) - I_1(\boldsymbol{q}_j), \tag{3}$$

where $d_j \in \mathbb{D} = \{d_j | j = 1 \cdots N_j\}$ is the depth of a pixel $\boldsymbol{q}_j$ at the image $I_1$, and $d_j \cdot \boldsymbol{q}_j$ upgrade the pixel $\boldsymbol{q}_j$ to its 3D coordinate. Thus, the optimization parameter is $\mathcal{X} = [\boldsymbol{T}_1, \boldsymbol{T}_2 \cdots \boldsymbol{T}_{N_i}, d_1, d_2 \cdots d_{N_j}]^\top$. The direct methods have the advantages of using all pixels with sufficient gradient magnitude. They have demonstrated superior performance, especially at less textured scenes. However, these methods also have some drawbacks:

- They are sensitive to initialization as demonstrated in (Mur-Artal et al., 2015) and (Tang et al., 2017) because the photometric error increases the non-convexity (Engel et al., 2018).
- They are sensitive to camera exposure and white balance changes. An automatic photometric calibration is required (Engel et al., 2018; 2016).
- They are more sensitive to outliers such as moving objects.

## 4 THE BA-NET ARCHITECTURE

To deal with the above challenges, we propose a feature-metric BA algorithm which estimates the same scene depth and camera motion parameters $\mathcal{X}$ as in photometric BA, but minimizes the feature-metric difference of aligned pixels:

$$e_{i,j}^f(\mathcal{X}) = F_i(\pi(\boldsymbol{T}_i, d_j \cdot \boldsymbol{q}_j)) - F_1(\boldsymbol{q}_j), \tag{4}$$

where $\mathbb{F} = \{F_i | i = 1 \cdots N_i\}$ are feature pyramids of images $\mathbb{I} = \{I_i | i = 1 \cdots N_i\}$. Similar to the photometric BA, our feature-metric BA considers more pixels than corners or blobs. It has the potential to learn more suitable features for SfM to deal with exposure changes, moving objects, etc.

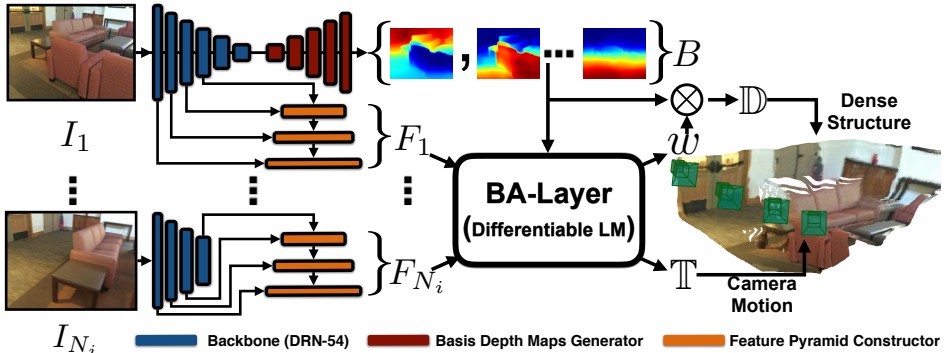

Figure 1: Overview of our BA-Net structure, which consists of a DRN-54 (Yu et al., 2017) as the backbone network, a **Basis Depth Maps Generator** that generates a set of basis depth maps, a **Feature Pyramid Constructor** that constructs multi-scale feature maps, and a **BA-Layer** that optimizes both the depth map and the camera poses through a novel differentiable LM algorithm.

We learn features suitable for SfM via back-propagation, instead of using pre-trained CNN features for image classification (Czarnowski et al., 2017). Therefore, it is crucial to design a differentiable optimization layer, our BA-Layer, to solve the optimization problem, so that the loss information can be back-propagated. The BA-Layer predicts the camera poses $\mathbb{T}$ and the dense depth map $\mathbb{D}$ during forward pass and back-propagates the loss from $\mathbb{T}$ and $\mathbb{D}$ to the feature pyramids $\mathbb{F}$ for training.

### 4.1 OVERVIEW

As illustrated in Figure 1, our BA-Net receives multiple images and then feed them to the backbone DRN-54. We use DRN-54 (Yu et al., 2017) because it replaces max-pooling with convolution layers and generates smoother feature maps, which is desirable for BA optimization. Note the original DRN is memory inefficient due to the high resolution feature maps after dilation convolutions. We replace the dilation convolution with ordinary convolution with strides to address this issue. After DRN-54, a feature pyramid is then constructed for each input image, which are the inputs for the BA-Layer.

At the same time, the basis depth maps generator generates multiple basis depth maps for the image $I_1$, and the final depth map is represented as a linear combination of these basis depth maps.

Finally, the BA-Layer optimizes for the camera poses and the dense depth map jointly by minimizing the feature-metric error defined in Equation (4), which makes the whole pipeline end-to-end trainable.

### 4.2 FEATURE PYRAMID

The feature pyramid learns suitable features for the BA-Layer. Similar to the feature pyramid networks (FPN) for object detection (Lin et al., 2017), we exploit the inherent multi-scale hierarchy of deep convolutional networks to construct feature pyramids. A top-down architecture with lateral connections is applied to propagate richer context information from coarser scales to finer scales. Thus, our feature-metric BA will have a larger convergence radius.

As shown in Figure 2(a), we construct a feature pyramid from the backbone DRN-54. We denote the last residual blocks of conv1, conv2, conv3, conv4 in DRN-54 as $\{C^1, C^2, C^3, C^4\}$, with strides $\{1, 2, 4, 8\}$ respectively. We upsample a feature map $C^{k+1}$ by a factor of 2 with bilinear interpolation and concatenate the upsampled feature map with $C^k$ in the next level. This procedure is iterated until the finest level. Finally, we apply a $3 \times 3$ convolution on the concatenated feature maps to reduce its dimensionality to 128 to balance the expressiveness and computational complexity, which leads to the final feature pyramid $F_i = [F_i^1, F_i^2, F_i^3]$ for image $I_i$.

We visualize some typical channels from the raw image $I$ (i.e. the RGB channels), the pre-trained DRN-54 $C^3$ and our learned $F^3$ in Figure 2(b). It is evident that, after training with our BA-Layer, the feature pyramid becomes smoother and each channel correspondences to different regions in the image. Note that our feature pyramids have higher resolution than FPN to facilitate precise alignment.

To have a better intuition about how much the BA optimization benefits from our learned features, we visualize different distances in Figure 3. We evaluate the distance between a pixel marked by a yellow cross in the top image in Figure 3 (a) and all pixels in a neighbourhood of its corresponding point in the bottom image of Figure 3 (a). The distances evaluated from raw RGB values, pretrained feature $C^3$, and our learned feature $F^3$ are visualized in (b), (c), and (d) respectively. All distances are normalized to $[0, 1]$ and visualized as heat maps. The $x$-axis and $y$-axis are the offsets to the ground-truth corresponding point. The RGB distance in (b) (i.e. $e^p$ in Equation (3)) has no clear global minimum, which makes the photometric BA sensitive to initialization (Engel et al., 2014; 2018). The distance measured by the pretrained feature $C^3$ has both global and local minimums. Finally, the distance measured by our learned feature $F^3$ has a clear global minimum and smooth basin, which is helpful in gradient based optimization such as the LM algorithm.

### 4.3 BUNDLE ADJUSTMENT LAYER

After building feature pyramids for all images, we optimize camera poses and a dense depth map by minimizing the feature-metric error in Equation (4). Following the conventional Bundle Adjustment principle, we optimize Equation (4) using the Levenberg-Marquardt (LM) algorithm. However, the original LM algorithm is non-differentiable because of two difficulties:

- The iterative computation terminates when a specified convergence threshold is reached. This if-else based termination strategy makes the output solution $\mathcal{X}$ non-differentiable with respect to the input $\mathbb{F}$ (Domke, 2012).

- In each iteration, it updates the damping factor $\lambda$ based on the current value of the objective function. It raises $\lambda$ if a step fails to reduce the objective; otherwise it reduces $\lambda$. This if-else decision also makes $\mathcal{X}$ non-differentiable with respect to $\mathbb{F}$.

When the solution $\mathcal{X}$ is non-differentiable with respect to $\mathbb{F}$, feature learning by back-propagation becomes impossible. The first difficulty has been studied in Domke (2012) and the author proposes to fix the number of iterations, which is refered as 'incomplete optimization'. Besides making the optimization differentiable, this 'incomplete optimization' technique also reduces memory consumption because the number of iterations is usually fixed at a small value.

The second difficulty has never been studied. Previous works mainly focus on gradient descent (Domke, 2012) or quadratic minimization (Amos & Kolter, 2017; Schmidt & Roth, 2014). In this section, we propose a simple yet effective approach to soften the if-else decision and yields a differentiable LM algorithm. We send the current objective value to a MLP network to predict $\lambda$. This technique not only makes the optimization differentiable, but also learns to predict a better damping factor $\lambda$, which helps the optimization to reach a better solution within limited iterations.

To start with, we illustrate a single iteration of the LM optimization as a diagram in Figure 4 by interpreting intermediate variables as network nodes. During the *forward* pass, we compute the solution update $\Delta \mathcal{X}$ from feature pyramids $\mathbb{F}$ and current solution $\mathcal{X}$ as the following steps:

- We compute the feature-metric error $E(\mathcal{X}) = [e_{1,1}^f(\mathcal{X}), e_{1,2}^f(\mathcal{X}) \cdots e_{N_i,N_j}^f(\mathcal{X})]$ with Equation (4) on all $N_i$ images and $N_j$ pixels, where $\mathcal{X}$ is the solution from the previous iteration;

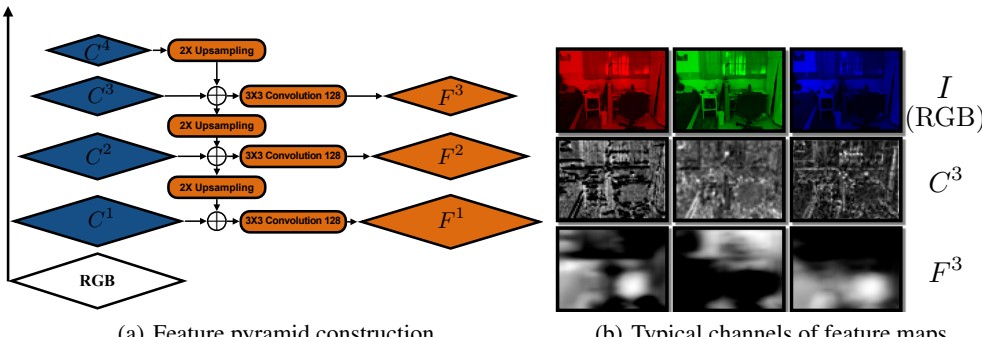

(a) Feature pyramid construction   (b) Typical channels of feature maps

Figure 2: A feature pyramid and some typical channels from different feature maps.

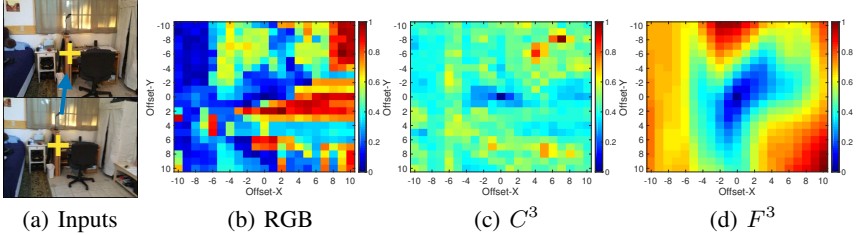

| (a) Inputs | (b) RGB | (c) $C^3$ | (d) $F^3$ |

Figure 3: Feature distance maps defined over raw RGB values, pretrained CNN features $C^3$, or our learned features $F^3$. Our features produce smoother objective function to facilitate optimization.

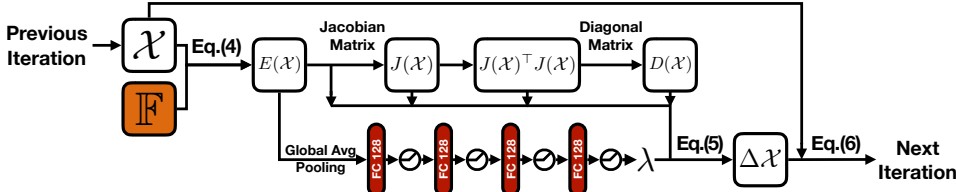

Figure 4: A single iteration of the differentiable LM.

- We then compute the Jacobian matrix $J(\mathcal{X})$, the Hessian matrix $J(\mathcal{X})^\top J(\mathcal{X})$ and its diagonal matrix $D(\mathcal{X})$;
- To predict the damping factor $\lambda$, we use global average pooling to aggregate the aboslute value of $E(\mathcal{X})$ over all pixels for each feature channel, and get a 128D feature vector. We then send it to a MLP sub-network to predict $\lambda$;
- Finally, the update $\Delta\mathcal{X}$ to the current solution is computed as a standard LM step:

$$\Delta\mathcal{X} = (J(\mathcal{X})^\top J(\mathcal{X}) + \lambda D(\mathcal{X}))^{-1} J(\mathcal{X})^\top E(\mathcal{X}). \tag{5}$$

In this way, we can consider $\lambda$ as an intermediate variable and denote each LM step as a function $g$ about features pyramids $\mathbb{F}$ and the solution $\mathcal{X}$ from the previous iteration. In other words, $\Delta\mathcal{X} = g(\mathcal{X}; \mathbb{F})$. Therefore, the solution after the $k$-th iteration is:

$$\mathcal{X}_k = g(\mathcal{X}_{k-1}; \mathbb{F}) \circ \mathcal{X}_{k-1}. \tag{6}$$

Here, $\circ$ denotes parameters updating, which is addition for depth and SE(3) exponential mapping for camera poses. Equation (6) is differentiable with respect to the feature pyramids $\mathbb{F}$, which makes back-propagation possible through the whole pipeline for feature learning. The MLP that predicts $\lambda$ is also shown in Figure 4. We stack four fully-connected layers to predict $\lambda$ from the input 128D vector. We use ReLU as the activation function to guarantee $\lambda$ is non-negative. Following the photometric BA (Engel et al., 2014; 2018), we solve our feature-metric BA using a coarse-to-fine strategy with feature map warping at each iteration. We apply the differentiable LM algorithm for 5 iterations at each pyramid level, leading to 15 iterations in total. All the camera poses are initialized with identity rotation and zero translation, and the initialization of depth map will be introduced in Section 4.4.

## 4.4 Basis Depth Maps Generation

Parameterizing a dense depth map by a per-pixel depth value is impractical under our formulation. Firstly, it introduces too many parameters for optimization. For example, an image of $320 \times 240$ pixels results in 76.8k parameters. Secondly, in the beginning of training, many pixels will become invisible in the other views because of the poorly predicted depth or motion. So little information can be back-propagated to improve the network, which makes training difficult.

To deal with these problems, we use the convolutional network for monocular image depth estimation as a compact parameterization, rather than using it as an initialization as in Tateno et al. (2017) and Yang et al. (2018). We use a standard encoder-decoder architecture for monocular depth learning as in Laina et al. (2016). We use DRN-54 as the encoder to share the same backbone features with our feature pyramids. For the decoder, we modify the last convolutional feature maps of Laina et al. (2016) to 128 channels and use these feature maps as the basis depth maps for optimization. The final depth map is generated as the linear combination of these basis depth maps, which is:

$$\mathbb{D} = \mathrm{ReLU}(\boldsymbol{w}^\top \boldsymbol{B}). \tag{7}$$

Here, $\mathbb{D}$ is the $h \cdot w$ depth map that contains depth values for all pixels, $\boldsymbol{B}$ is a $128 \times h \cdot w$ matrix, representing 128 basis depth maps generated from network, $\boldsymbol{w}$ is the linear combination weights of these basis depth maps. The $\boldsymbol{w}$ will be optimized in our BA-Layer. The ReLU activation function guarantees the final depth is non-negative. Once $\boldsymbol{B}$ is generated from the network, we fix $\boldsymbol{B}$ and use $\boldsymbol{w}$ as a compact depth parameterization in BA optimization, and the feature-metric distance becomes:

$$e_{i,j}^{f}(\mathcal{X}) = F_i(\pi(\boldsymbol{T}_i, \mathrm{ReLU}(\boldsymbol{w}^\top \boldsymbol{B}[j]) \cdot \boldsymbol{q}_j)) - F_1(\boldsymbol{q}_j), \qquad (8)$$

where $\boldsymbol{B}[j]$ is the $j$-th column of $\boldsymbol{B}$, and $\mathrm{ReLU}(\boldsymbol{w}^\top \boldsymbol{B}[j])$ is the corresponding depth of $\boldsymbol{q}_j$. To further speedup convergence, we learn the initial weight $\boldsymbol{w}_0$ as a 1D convolution filter for an arbitrary image, i.e. $\mathbb{D}_0 = \mathrm{ReLU}(\boldsymbol{w}_0^\top \boldsymbol{B})$. The $\boldsymbol{B}$ of various images are visualized in the appendix.

### 4.5 TRAINING

The BA-Net learns the feature pyramid, the damping factor predictor, and the basis depth maps generator in a supervised manner. We apply the following commonly used loss for training, though more sophisticated ones might be designed.

**Camera Pose Loss** The camera rotation loss is the distance between rotation quaternion vectors $\mathcal{L}_{rotation} = \|\boldsymbol{q} - \boldsymbol{q}^*\|$. Similarly, translation loss is the Euclidean distance between prediction and groundtruth in metric scale, $\mathcal{L}_{translation} = \|\boldsymbol{t} - \boldsymbol{t}^*\|$.

**Depth Map Loss** For each dense depth map we applies the berHu Loss (Zwald & Lambert-Lacroix, 2012) as in Laina et al. (2016).

We initialize the back-bone network from DRN-54 (Yu et al., 2017), and the other components are trained with ADAM (Kingma & Ba, 2015) from scratch with initial learning rate 0.001, and the learning rate is divided by two when we observe plateaus from the Tensorboard interface.

## 5 EVALUATION

### 5.1 DATASET

**ScanNet** ScanNet (Dai et al., 2017a) is a large-scale indoor dataset with 1,513 sequences in 706 different scenes. Camera poses and depth maps are not perfect, because they are estimated via BundleFusion (Dai et al., 2017b). The metric scale is known in all data from ScanNet, because the data are recorded with a depth camera which returns absolute depth values.

To sample image pairs for training, we apply a simple filtering process. We first filter out pairs with a large photo-consistency error, to avoid image pairs with large pose or depth error. We also filter out image pairs, if less than 50% of the pixels from one image are visible in the other image. In addition, we also discard a pair if their roundness score (Beder & Steffen, 2006) is less than 0.001, which avoids pairs with too narrow baselines.

We split the whole dataset into the training and the testing sets. The training set contains the first 1,413 sequences and the testing set contains the rest 100 sequences. We sample 547,991 training pairs and 2,000 testing pairs from the training and testing sequences respectively.

**KITTI** KITTI (Geiger et al., 2012) is a widely used benchmark dataset collected by car-mounted cameras and a LIDAR sensor on streets. It contains 61 scenes belonging to the "city", "residential", or "road" categories. Eigen et al. (2014) select 28 scenes for testing and 28 scenes from the remaining for training. We use the same data split, to make a fair comparison with previous methods. Since ground truth pose is unavailable from the raw KITTI dataset, we compute camera poses by LibVISO2 (Geiger et al., 2011) and take them as ground truth after discarding poses with large errors.

### 5.2 COMPARISONS WITH OTHER METHODS

**ScanNet** To evaluate the results' quality, we use the depth error metrics suggested in Eigen & Fergus (2015), where RMSE (linear, log, and log, scale inv.) measure the RMSE of the raw, the logarithmical, and aligned logarithmical depth values, while the other two metrics measure the mean of the ratios that divide the absolute and square error by groundtruth depth.. The errors in camera

|  | Ours | Ours* | DeMoN* | Photometric BA | Geometric BA |
|---|---|---|---|---|---|
| Rotation (degree) | **1.018** | 1.587 | 3.791 | 4.409 | 8.56 |
| Translation (cm) | **3.39** | 10.81 | 15.5 | 21.40 | 36.995 |
| Translation (degree) | **20.577** | 31.005 | 31.626 | 34.36 | 39.392 |
| abs relative difference | **0.161** | 0.238 | 0.231 | 0.268 | 0.382 |
| sqr relative difference | **0.092** | 0.176 | 0.520 | 0.427 | 1.163 |
| RMSE (linear) | **0.346** | 0.488 | 0.761 | 0.788 | 0.876 |
| RMSE (log) | **0.214** | 0.279 | 0.289 | 0.330 | 0.366 |
| RMSE (log, scale inv.) | **0.184** | 0.276 | 0.284 | 0.323 | 0.357 |

Table 1: Quantitative comparisons with DeMoN and classic BA. The superindex * denotes that the model is trained on the trainning set described in Ummenhofer et al. (2017).

poses are measured by the rotation error (the angle between the ground truth and the estimated camera rotations), the translation direction error (the angle between the ground truth and estimated camera translation directions) and the absolute position error (the distance between the ground truth and the estimated camera translation vectors).

In Table 1, we compare our method with DeMoN (Ummenhofer et al., 2017) and the conventional photometric and geometric BA. Note that we cannot get DeMoN trained on the ScanNet. For fair comparison, we train our network on the same training data as DeMoN and test both networks on our testing data[1]. We also show the results of our network trained on ScanNet. Our BA-Net consistently performs better than DeMoN no matter which training data is used. Since DeMoN does not recover the absolute scale, we align its depth map with the groundtruth to recover its metric scale for evaluation. We further compare with conventional geometric (Nister, 2004; Agarwal et al.) and photometric (Engel et al., 2014) BA. Again, our method produces better results. The geometric BA works poorly here, because feature matching is difficult in indoor scenes. Even the RANSAC process cannot get rid of all outliers. While for photometirc BA, the highly non-convex objective function is difficult to optimize as described in Section 3.

**KITTI** We use the same metrics as the comparisons on ScanNet for depth evaluation. To evaluate the camera poses, we follow (Zhou et al., 2017; Wang et al., 2018) to use the Absolute Trajectory Error (ATE), which measures the Euclidean differences between two trajectories (Steinbruecker et al., 2011), on the 9th and 10th sequences from the KITTI odometry data. In this experiment, we create short sequences of 5 frames by first computing 5 two-view reconstructions from our BA-Net and then align the two-view reconstructions in the coordinate system anchored at the first frame. minimize the photometric error.

|  | Ours | Wang et al. (2018) | Zhou et al. (2017) | Godard et al. (2017) | Eigen et al. (2014) |
|---|---|---|---|---|---|
| ATE(km) | **0.019** | 0.045 | 0.021 | N/A | N/A |
| abs rel | **0.083** | 0.151 | 0.208 | 0.148 | 0.203 |
| sqr rel | **0.025** | 1.257 | 1.768 | 1.344 | 1.548 |
| RMSE(linear) | **3.640** | 5.583 | 6.856 | 5.927 | 6.307 |
| RMSE(log) | **0.134** | 0.228 | 0.283 | 0.247 | 0.282 |

Table 2: Quantitative comparisons on KITTI with supervised (Eigen et al., 2014) and unsupervised (Wang et al., 2018; Zhou et al., 2017; Godard et al., 2017) methods.

Table 2 summarizes our results on KITTI. Our method outperforms the supervised methods (Eigen et al., 2014) as well as recent unsupervised methods (Zhou et al., 2017; Wang et al., 2018; Godard et al., 2017). Our method also achieves more accurate camera trajectories than Zhou et al. (2017) and Wang et al. (2018). We believe this is due to our feature-metric BA with features learned specifically for SfM problem, which makes the objective function closer to convex and easier to optimize as discussed in Section 4.2. In comparison, Zhou et al. (2017) and Wang et al. (2018) minimize the photometric error.

More comparison with DeMoN, ablation studies, and multi-view SfM (up to 5 views) are reported in the appendix due to page limit.

---

[1]More comparison with DeMoN on DeMoN's data is provided in the appendix.

## 6 Conclusions and Future Works

This paper presents the BA-Net, a network that explicitly enforces multi-view geometry constraints in terms of feature-metric error. It optimizes scene depths and camera motion jointly via feature-metric bundle adjustment. The whole pipeline is differentiable and thus end-to-end trainable, such that the features are learned from data to facilitate structure-from-motion. The dense depth is parameterized as a linear combination of several basis depth maps generated from the network. Our BA-Net nicely combines domain knowledge (hard-coded multi-view geometry constraint) with deep learning (learned feature representation and basis depth maps generator). It outperforms conventional BA and recent deep learning based methods.

**Acknowledgement** This work is supported by the NSERC discovery grant 611664 and a project funding from Alibaba.

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

## APPENDIX A: IMPLEMENTATION DETAILS

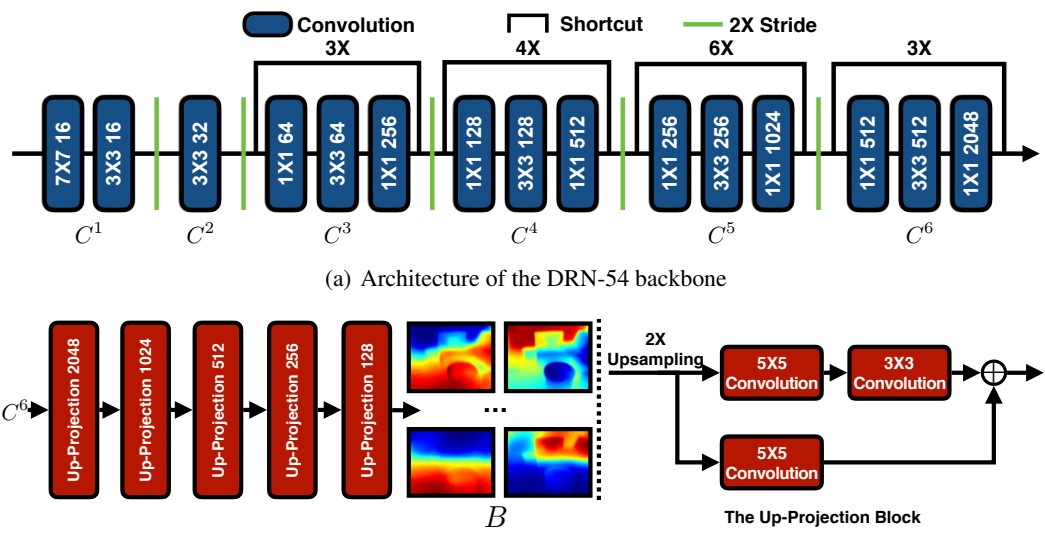

(a) Architecture of the DRN-54 backbone

(b) Architecture of the basis depth generator

Figure 5: Network details for the (a) the DRN-54 backbone and (b) the basis depth generator.

**Network Architecture Details** Figure 5 illustrates the detailed network architectures for the backbone DRN-54 and the depth basis generator. The architecture of the feature pyramid has been provided in Figure 2(a). We modify the dilated convolution of the original DRN-54 to convolution with strides and discard the conv7 and conv8 as shown in Figure 5(a). $C^1$ to $C^6$ are layers with $\{1,2,4,8,16,32\}$ strides and $\{16,32,256,512,1024,2048\}$ channels, where $C^1$ and $C^2$ are basic convolution layers, while $C^3$ to $C^6$ are standerd bottleneck blocks as in ResNet (He et al., 2016).

Figure 5(b) visualizes our depth basis generator which adopts the up-projection structure proposed in Laina et al. (2016). The depth basis generator is a stander decoder that takes the output of $C^6$ as input and stacks five up-projection blocks to generate 128 basis depth maps, and each of the basis depth maps is half the resolution of the input image. The up-projection block is shown on the right of Figure 5(b) which upsample the input by $2\times$ and then apply convolutions with projection connection.

**Evaluation Time** To evaluate the running time of our method, we use the Tensorflow profiler tool to retrieve the time in ms for all network nodes and then summarize the results corresponding to each component in our pipeline. As shown in Table 3, our method takes 95.21 ms to reconstruct two $320 \times 240$ images, which is slightly faster than DeMoN that takes 110 ms for two $256 \times 192$ images.

The current computation bottleneck is the BA-Layer which contains a large amount of matrix operations and can be further speeded up by direct CUDA implementation. Since we explicitly hard-code the multi-view geometry constraints in the BA-Layer, it is possible to share the backbone DRN-54 with other high-level vision tasks, such as semantic segmentation and object detection, to maximize reuse of network structures and minimize extra computation cost.

|  | Backbone (DRN-54) | Feature Pyramid | Basis Depth Generator | BA-Layer Optimization | Total |
|---|---|---|---|---|---|
| Time (ms) | 15.04 | 5.87 | 9.81 | 67.22 | 95.21 |

Table 3: Evaluation time for each component, which is summarized using Tensorflow profiler.

## APPENDIX B: ABLATION STUDIES

**Learned Features vs Pre-trained Features** Our learned feature pyramid improves the convexity of the objective function to facilitate the optimization. We compare our learned features with features

|  | Ours (Full) | w/o Feature Learning | w/o Joint Optimization | w/o $\lambda$ |
|---|---|---|---|---|
| Rotation (degree) | **1.018** | 2.667 | 1.036 | 7.202 |
| Translation (cm) | **3.39** | 10.8 | 3.91 | 22.38 |
| Translation (degree) | **20.577** | 31.493 | 26.779 | 59.81 |
| abs relative difference | **0.161** | 0.267 | 0.217 | 0.630 |
| sqr relative difference | **0.092** | 0.242 | 0.145 | 0.549 |
| RMSE (linear) | **0.346** | 0.481 | 0.428 | 0.763 |
| RMSE (log) | **0.214** | 0.303 | 0.270 | 0.513 |
| RMSE (log, scale inv.) | **0.184** | 0.226 | 0.205 | 0.437 |

Table 4: Ablation Study Comparisons by Disabling Different Components of BA-Net

pre-trained on ImageNet for classification tasks. As shown in Table 4, the pre-trained features (i.e. w/o Feature Learning) produce larger error. This proves the discussion in Section 4.2.

**Bundle Adjustment Optimization vs SE(3) Pose Estimation**  Our BA-Layer optimizes depth and camera poses jointly. We compare it to the SE(3) camera pose estimation with fixed depth map (e.g. the initialized depth $\mathbb{D}_0$ in Section 4.4), and similar strategy is adopted in Wang et al. (2018). To make a fair comparison, we also use our learned feature pyramids for the SE(3) camera pose estimation. As shown in Table 4, without BA optimization (i.e. w/o Joint Optimization), both the depth maps and camera poses are worse, because the errors in the depth estimation will degrades the camera pose estimation.

**Differentiable Levenberg-Marquardt vs Gauss-Newton**  To make the whole pipeline end-to-end trainable, we makes the Levenberg-Marquardt algorithm differentiable by learning the damping factor from the network. We first compare our method against vanilla Gauss-Newton without damping factor $\lambda$ (i.e. $\lambda = 0$). Since the objective function of feature-metric BA is non-convex, the Hessian matrix $J(\mathcal{X})^\top J(\mathcal{X})$ might not be positive definite, which makes the matrix inversion by Cholesky decomposition fail.

To deal with this problem, we use QR decomposition instead for training with Gauss-Newton. As shown in Table 4, the Gauss-Newton algorithm (i.e. w/o $\lambda$) generates much larger error, because the BA optimization is non-convex and the Gauss-Newton algorithm has no guaranteed convergence unless the initial solution is sufficiently close to the optimal (Nocedal & Wright, 2006). This comparison reveals that, similar to conventional BA, our differnetiable Levenberg-Marquardt algorithm is superior than the Gauss-Newton algorithm for feature-metric BA.

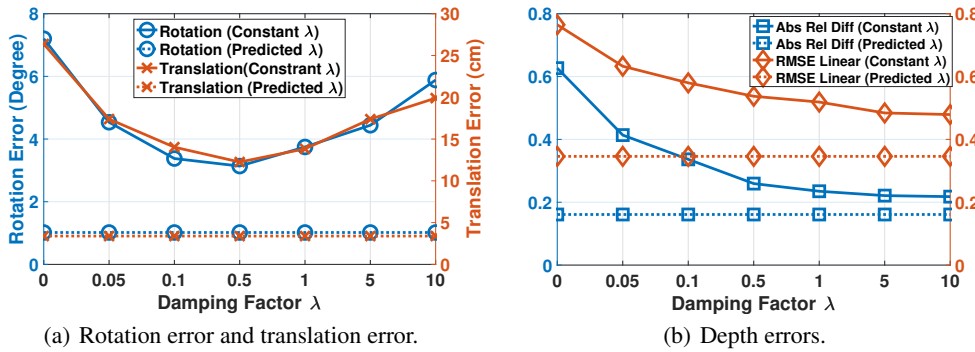

(a) Rotation error and translation error.    (b) Depth errors.

Figure 6: The camera pose and the depth errors correspond to different constant $\lambda$ values.

**Predicted vs Constant $\lambda$**  Another way to make the Levenberg-Marquardt algorithm differentiable is to fix the $\lambda$ during the iterations. We compare with this strategy. As shown in Figure 6(a), increasing $\lambda$ makes the both rotation and translation error decreases, until $\lambda = 0.5$, and then increases. The reason is that a small $\lambda$ makes the algorithm close to the Gauss-Newton algorithm, which has convergence issues. A large $\lambda$ leads to a small update at each iteration, which makes it difficult to reach a good solution within limited iterations.

While in Figure 6(b), increasing $\lambda$ always makes depth errors decrease, probably because a larger $\lambda$ leads to a small update and makes the final depth close to the initialed depth, which is better than the optimized one with small constant $\lambda$.

Using constant $\lambda$ value consistently generates worse results than using a predicted $\lambda$ from the MLP network, because there is no optimal $\lambda$ for all data and it should be adapted to different data and different iterations. We draw the errors of our method in Figure 6(a) and Figure 6(b) as the flat dash lines for a reference.

## APPENDIX C: EVALUATION ON DEMON DATASET

Table 5 summarizes our results on the DeMoN dataset. For a comparison, we also cite the results from DeMoN (Ummenhofer et al., 2017) and the most recent work LS-Net (Clark et al., 2018). We further cite the results from some conventional approaches as reported in DeMoN, indicated as **Oracle**, **SIFT**, **FF**, and **Matlab** respectively. Here, **Oracle** uses *ground truth* camera poses to solve the multi-view stereo by SGM (Hirschmuller, 2005), while **SIFT**, **FF**, and **Matlab** further use sparse features, optical flow, and KLT tracking respectively for feature correspondence to solve camera poses by the 8-pt algorithm (Hartley, 1997).

|  | | Depth | | | Motion | |  | | | Depth | | | Motion | |
| --- | --- | --- | --- | --- | --- | --- | --- | --- | --- | --- | --- | --- | --- | --- |
|  | Method | L1-inv | sc-inv | L1-rel | Rotation | Translation | | | Method | L1-inv | sc-inv | L1-rel | Rotation | Translation |
| MVS | Oracle | 0.019 | 0.197 | 0.105 | 0 | 0 | | | Oracle | 0.023 | 0.618 | 0.349 | 0 | 0 |
| | SIFT | 0.056 | 0.309 | 0.361 | 21.180 | 60.516 | Scenes11 | | SIFT | 0.051 | 0.900 | 1.027 | 6.179 | 56.650 |
| | FF | 0.055 | 0.308 | 0.322 | 4.834 | 17.252 | | | FF | 0.038 | 0.793 | 0.776 | 1.309 | 19.425 |
| | Matlab | - | - | - | 10.843 | 32.736 | | | Matlab | - | - | - | 0.917 | 14.639 |
| | DeMoN | 0.047 | 0.202 | 0.305 | 5.156 | 14.447 | | | DeMoN | 0.019 | 0.315 | 0.248 | **0.809** | 8.918 |
| | LS-Net | 0.051 | 0.221 | 0.311 | 4.653 | **11.221** | | | LS-Net | **0.010** | 0.410 | 0.210 | 0.910 | **8.21** |
| | Ours | **0.03** | **0.15** | **0.08** | **3.499** | 11.238 | | | Ours | 0.08 | **0.21** | **0.13** | 1.298 | 10.37 |
|  | | Depth | | | Motion | |  | | | Depth | | | Motion | |
| | Method | L1-inv | sc-inv | L1-rel | Rotation | Translation | | | Method | L1-inv | sc-inv | L1-rel | Rotation | Translation |
| RGB-D | Oracle | 0.026 | 0.398 | 0.336 | 0 | 0 | | | Oracle | 0.020 | 0.241 | 0.220 | 0 | 0 |
| | SIFT | 0.050 | 0.577 | 0.703 | 12.010 | 56.021 | Sun3D | | SIFT | 0.029 | 0.290 | 0.286 | 7.702 | 41.825 |
| | FF | 0.045 | 0.548 | 0.613 | 4.709 | 46.058 | | | FF | 0.029 | 0.284 | 0.297 | 3.681 | 33.301 |
| | Matlab | - | - | - | 12.831 | 49.612 | | | Matlab | - | - | - | 5.920 | 32.298 |
| | DeMoN | 0.028 | 0.130 | 0.212 | 2.641 | 20.585 | | | DeMoN | 0.019 | 0.114 | 0.172 | 1.801 | 18.811 |
| | LS-Net | 0.019 | 0.09 | 0.301 | **1.01** | 22.1 | | | LS-Net | **0.015** | 0.189 | 0.650 | **1.521** | 14.347 |
| | Ours | **0.008** | **0.087** | **0.05** | 2.459 | **14.90** | | | Ours | **0.015** | **0.11** | **0.06** | 1.729 | **13.26** |

Table 5: Quantitative comparisons on the DeMoN dataset.

Our method consistently outperforms DeMoN (Ummenhofer et al., 2017) at both camera motion and scene depth, except on the 'Scenes11' data, because we enforce multi-view geometry constraint in the BA-Layer. Our results are poorer on the 'Scene11' dataset, because the images there are synthesized with random objects from the ShapeNet (Chang et al., 2015) without physically correct scale. This setting is inconsistent with real data and makes it harder for our method to learn the basis depth map generator.

When compared with LS-Net Clark et al. (2018), our method achieves similar accuracy on camera poses but better scene depth. It proves our feature-metric BA with learned feature is superior than the photometric BA in the LS-Net.

## APPENDIX D: MULTI-VIEW STRUCTURE-FROM-MOTION

Our method can be easily extended to reconstruct multiple images. We evaluate our method in the multi-view setting on the ScanNet (Dai et al., 2017a) dataset. To sample multi-view images for training, we randomly select two-view image pairs that shares a common image to construct $N$-view sequences. Due to the limited GPU memory (12G), we limit $N$ to 5.

As shown in the Table 6, the accuracy is consistently improved when more views are included, which demonstrates the strength of the multi-view geometry constraints. Instead, most existing deep learning approaches can only handle two views at a time, which is sub-optimal as known in structure-from-motion literature.

|  | Ours(2-views) | Ours(3-views) | Ours(5-views) |
|---|---|---|---|
| Rotation (degree) | 1.018 | 1.013 | **1.009** |
| Translation (cm) | 3.391 | 2.852 | **2.365** |
| Translation (degree) | 20.577 | 16.423 | **14.626** |
| abs relative difference | 0.161 | 0.111 | **0.091** |
| sqr relative difference | 0.092 | 0.087 | **0.068** |
| RMSE (linear) | 0.346 | 0.288 | **0.223** |
| RMSE (log) | 0.214 | 0.179 | **0.147** |
| RMSE (log, scale inv.) | 0.184 | 0.168 | **0.137** |

Table 6: Quantitative comparisons on multi-view reconstruction on ScanNet.

## APPENDIX E: QUANTITATIVE COMPARISONS WITH CODESLAM

We compare our method with CodeSLAM (Bloesch et al., 2018) which adopts similar idea for depth parameterization. But the difference is that CodeSLAM learns the conditioned depth auto-encoder separately and uses the depth codes in a standalone photometric BA component, while our method learns the feature pyramid and basis depth maps generator through feature-metric BA end-to-end. Since there is no public code for CodeSLAM, we directly cite the results from their paper.[2] To get the trajectory on the EuroC MH02 sequence of our method, we select one frame every four frames and concatenate the reconstructed groups that contains every five selected frames. Then we use the same evaluation metrics as in CodeSLAM, which measures the translation errors correspond to different traveled distances.

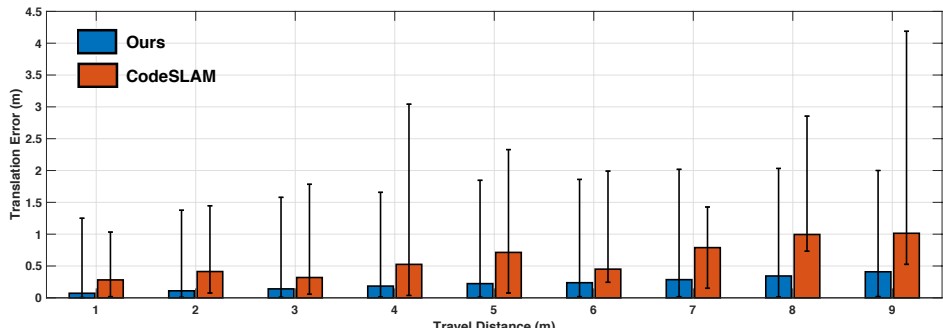

Figure 7: Quantitative Comparisons with CodeSLAM (Bloesch et al., 2018) on EuroC MH02. The error bars represent the maximum and the minimum errors. The orange and the blue boxes represent the median errors for CodeSLAM and our method.

As shown in Figure 7, our method outperforms CodeSLAM. Our median error is less than the half of CodeSLAM's error, i.e. CodeSLAM exhibits an error of roughly 1 m for a traveled distance of 9 m, while our method's error is about 0.4 m. This comparison demonstrates the superiority of end-to-end learning with feature pyramid and feature-metric BA over learning depth parameterization only.

## APPENDIX F: VISUALIZATION OF BASIS DEPTH MAPS

In Figure 8, we visualize four typical basis depth maps as heat maps for each of the four images. An interesting observation is that one basis depth map has higher responses on close objects while another oppositely has higher responses to the far background. Some other basis depth maps have smoothly varying responses and correspond to the layouts of scenes. This observation reveals that our learned basis depth maps have captured the latent structures of scenes.

---

[2]We thank the authors of CodeSLAM to share their source file of the figure in their paper.

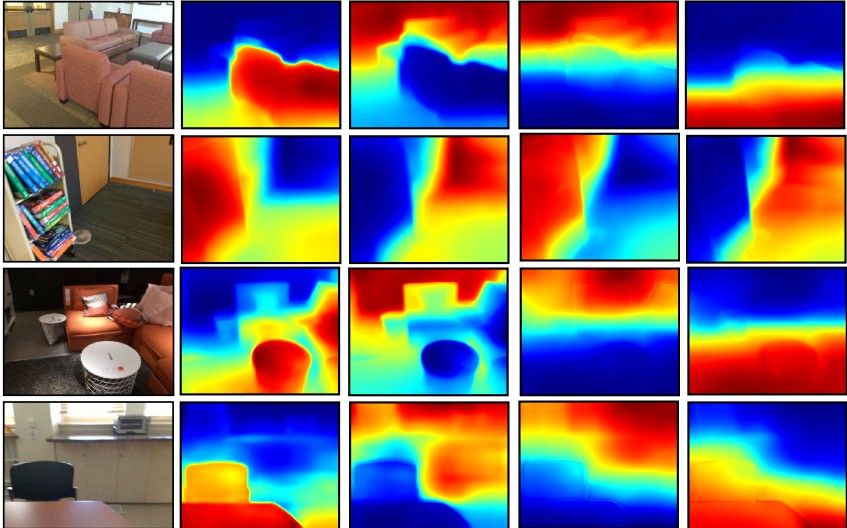

Figure 8: Visualization of different basis depth maps.

## APPENDIX G: QUALITATIVE COMPARISONS WITH OTHER METHODS

Finally, we show some qualitative comparison with the previous methods. Figure 9 shows the recovered depth map by our method and DeMoN Ummenhofer et al. (2017) on the ScanNet data. As we can see from the regions highlighted with a red circle, our method recovers more shape details. This is consistent with the quantitative results in Table 1. Figure 11 shows the recovered depth maps by our method, Wang et al. (2018), and Godard et al. (2017) respectively. Similarly, we observe more shape details in our results, as reflected in the quantitative results in Table 2.

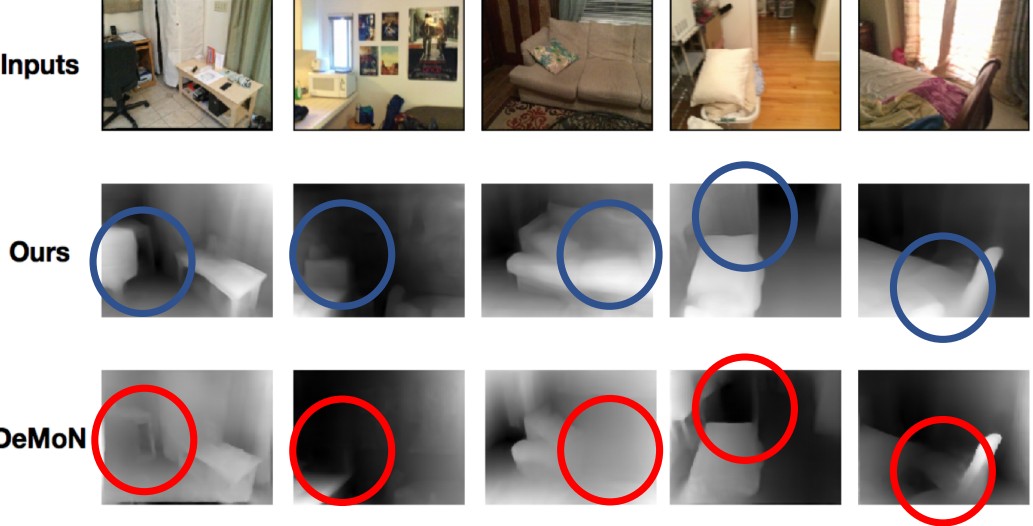

Figure 9: Qualitative Comparisons with DeMoN (Ummenhofer et al., 2017) on ScanNet.

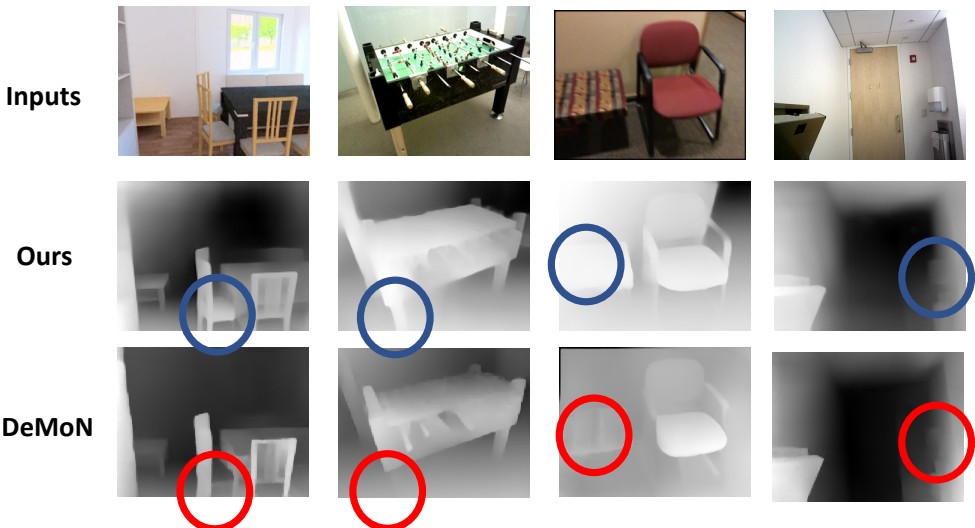

Figure 10: Qualitative Comparisons with DeMoN (Ummenhofer et al., 2017) on its dataset.

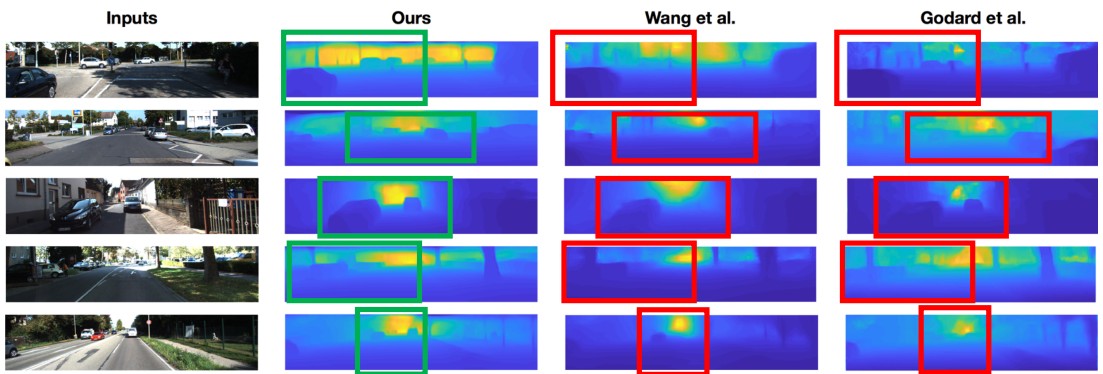

Figure 11: Qualitative Comparisons with Wang et al. (2018) and Godard et al. (2017).

