# OpenReview forum: "BA-Net: Dense Bundle Adjustment Networks"
_ICLR.cc/2019/Conference_

### Official Review · AnonReviewer2 · 2018-10-31
**dense SfM with Deep Learning**

**Rating:** 8
**Confidence:** 4

**Review:**

edit: the authors added several experiments (better evaluation of the predicted lambda, comparison with CodeSLAM), which address my concerns. I think the paper is much more convincing now. I am happy to increase my rating to clear accept.

I also agree with the introduction of the Chi vector, and with the use of the term of "photometric BA", since it was used before, even if it is unfortunate in my opinion. I thank the authors to replace reprojection by alignment, which is much clearer.

---------------


This paper presents a method for dense Structure-from-Motion using Deep Learning:
The input is a set of images; the output is the camera poses and the depth maps for all the images.
The approach is inspired by Levenberg-Marquardt optimization (LM): A pipeline extracting image features computes the Jacobian of an error function. This Jacobian is used to update an estimate of the camera poses. As in LM optimization, this update is done based on a factor lambda, weighting a gradient descent step and a Gauss-Newton step. In LM optimization, this lambda evolves with the improvement of the estimate. Here lambda is also predicted using a network based on the feature difference.

If I understand correctly, what is learned is how to compute image features that provide good updates, how to predict the depth maps from the features, and how to predict lambda.

The method is compared against DeMoN and other baselines with good results.

I like the fact that the method is based on LM optimization, which is the standard method in 'geometric bundle adjustment', while related works consider Gauss-Newton-like optimization steps. The key was to include a network to predict lambda as well.

However, I have several concerns:

* the ablation study designed to compare with a Gauss-Newton-like approach does not seem correct. The image features learned with the proposed method are re-used in an approach using a fixed lambda. If I understand correctly, there are 2 things wrong with that:
- for GN optimization, lambda should be set to 0 - not a constant value. Several constant values should also have been tried.
- the image features should be re-trained for the GN framework:  Since the features are learned for the LM iteration, they are adapted to the use of the predicted lambda, but they are not necessarily suitable to GN optimization.
Thus, the advantage of using a LM optimization scheme is not very convincing.

Since the LM-like approach is the main contribution, and the reported experiments do not show an advantage over GN-like approaches (already taken by previous work), this is my main reason for proposing rejection.

* CodeSLAM (best paper at CVPR'18) is referenced but there is no comparison with it, while a comparison on the EuRoC dataset should be possible.

Less critical concerns that still should be taken into account if the paper is accepted:

- the state vector Chi is not defined for the proposed method, only for the standard bundle adjustment approach. If I understand correctly is made of the camera poses.

- the name 'Bundle Adjustment' is actually not adapted to the proposed method.  'Bundle Adjustment' in 'geometric computer vision' comes from the optimization of several rays to intersect at the same 3D point, which is done by minimizing the reprojection errors. Here the objective function is based on image feature differences. I thus find the name misleading. The end of Section 3 also encourages the reader to think that the proposed method is based on the reprojection error. The proposed method is more about dense alignment for multiple images.


More minor points:

1st paragraph:  Marquet -> Marquardt
title of Section 3: revisitED
1st paragraph of Section 3: audience -> reader
caption of Fig 1: extractS
Eq (2) cannot have Delta Chi on the two sides. Typically, the left side should be \hat{\Delta \Chi}
before Eq (3): the 'photometric ..' -> a 'photometric ..'
1st paragraph of Section 4.3: difficulties -> reason
typo in absolute in caption of Fig 4
Eq (6): Is B the same for all scenes?  It would be interesting to visualize it.
Section 4.5: applies -> apply

---

> ### Author Response · Authors · 2018-11-27
> **Response to Reviewer 3**
>
> We thank the reviewer for the comments and appreciate that the reviewer likes our idea of including optimization in the network. But our contribution is beyond adopting Levenberg-Marquardt instead of Gauss-Newton. We would like to clarify several things to address the reviewer's concerns:
>
> Q1. The advantages of Levenberg-Marquardt over Gauss-Newton is unclear (the main reason for rejection):
>
> Firstly, we want to clarify that our contribution is beyond improving the Gauss-Newton optimization to Levenberg-Marquardt. More importantly, our contribution is the combination of conventional multi-view geometry (i.e. joint optimization of depth and camera poses) and end-to-end deep learning (I.e. depth basis generator learning and feature learning). This contribution is achieved by our differentiable LM optimization that allows end-to-end training.
>
> Secondly, we agree with the reviewer that comparing with the Gauss-Newton algorithm will be interesting and have updated such a comparison in Appendix B in the revised version according to the reviewer’s suggestions:
>
>     1. We retrained the whole pipeline with Gauss-Newton, to make sure the features are learned specifically for Gauss-Newton.
>
>     2. We compared with various constant lambda values to see how the performance varies along with lambda. Note that we also fine-tune the network to make sure the features fit different lambda.
>
> In Table 4 of the revised version (Appendix B), our method outperforms the Gauss-Newton algorithm in the last column. This is because the objective function to be optimized is non-convex, and the vanilla Gauss-Newton method might get stuck at saddle point or local minimum. This is why the Levenberg-Marquardt algorithm is the standard choice for conventional bundle adjustment.
>
> In Figure 6 of the revised version (Appendix B), our method also consistently performs better than different constant lambda values. This is because the value of lambda should be adapted to different data and optimization iterations. There is no ‘optimal’ constant lambda for all data and iterations.
>
>
> Q2.  Comparison with CodeSLAM:
> We have included that in Figure 7 of the revised version (Appendix E). Since there is no public code for CodeSLAM, we cite its results directly from the CodeSLAM paper.
>
> Q3. The state vector Chi is not defined for the proposed method.
> The Chi is defined in Section 3 as the vector containing all camera poses and point depths. Since our method also solves for these unknowns as in classic methods, we did not redefine the Chi. But in the revised version we have recapped the definition of Chi when introducing our method at the beginning of Section 4.
>
> Q4. Should the paper be called Bundle Adjustment?:
> The term ‘Bundle Adjustment’ is originally used to refer to the joint optimization of 3D scene points and camera poses by minimizing the reprojection error. The keyword Bundle comes from the fact that a bundle of camera view rays pass through each of the 3D scene points. Multiple recent works, e.g. [Engel et al., 2017,Delaunoy and Pollefeys, 2014], have generalized it to “photometric BA” where scene points and camera poses are optimized together by minimizing the photometric error. Our method is along this line. But we further improve the photometric error to featuremetric error. Each 3D scene point is still constrained by a bundle of camera view rays, though the error function has been changed. So we believe it is justified to call this method feature-metric BA.
>
> But we agree with the reviewer that the word ‘reprojection’ is misleading when we introduce our feature-metric BA and the photometric BA. So we use the word ‘align’ as the reviewer suggested and use ‘reprojection’ only for the geometric BA.
>
> Q5. Is B the same for all scenes?:
> In the revised version, We added Figure 8 to visualize of the term B in Equation 7 (Page 6) for different scenes. We can clearly see that it is scene dependent.
>
> Q6.Typos:
> We have fixed all the typos as suggested in the revised version.

---

> ### Author Response · Authors · 2018-11-27
> **We thank the reviewer for raising the score.**
>
> We thank the reviewer for raising the score.
>
> We submitted the response and the revision until the last minute because a lot of extra works have been done for the revision, and we want to ensure the correctness and completeness.
>
> But we will have a better-planned schedule for the next ICLR to fit the purpose of openreview.

---

### Official Review · AnonReviewer3 · 2018-10-31
**An interesting work but lacking some details concerning the implementation and experimentations**

**Rating:** 7
**Confidence:** 4

**Review:**

I believe that the authors have a solid contribution that can be interesting for the ICLR community.
Therefore, I recommend to accept the paper but after revision because the presentation and explanation of the ideas contain multiple typos and lacking some details (see bellow).

Summary:
The authors propose a new method called BA-Net to solve the SfM problem by explicitly incorporating geometry priors into a machine learning task. The authors focus on the Bundle Adjustment process.

Given several successive frames of a video sequence (2 frames but can be extended up to 5), BA-Net jointly estimates the depth of the first frame and the relative camera motion (between the first frame and the next one).
The method is based on a convolutional neural network which extracts the features of the different pyramid levels of the two images and in parallel computes the depth map of the first frame. The proposed network is based on the DRN-54 (Yu et al., 2017) as a feature extractor.

This is complemented by the linear combination of depth bases obtained from the first image.
The features and the initial depth then passed to the optimization layer called BA-layer where the feature re-projection error is minimized by the modified LM algorithm.

The authors adapt the standard multi-view geometry constraints by a new concept of feature re-projection error in the BA framework (BA-layer) which they made differentiable.
Differentiable optimization of camera motion and image depth via LM algorithm is now possible and can be used in various other DL architectures (ex. MVS-Net can probably benefit from BA-layer).

The authors also propose a novel depth parametrization in the form of linear combination of depth bases which reduces the number of parameters for the learning task,
enables integration into the same backbone net as used or feature pyramids and makes it possible to jointly train the depth generator and the BA-layer.

Originally the proposed approach depicts the network operating in the two-view settings. The extensibility to more views is also possible and, as shown by authors, proved to improve performance. It is, however, limited by the GPU capacity.

Overall, the authors came up with an interesting approach to the standard BA problem. They have managed to inject the multi-view geometry priors and BA into the DL architecture.

Major comments regarding the paper:

It would be interesting to know the evaluation times for the BA-net and more importantly to have some implementation details to ensure reproducibility.

Minor comments regarding the paper:

-	The spacing between sections is not consistent.
-	Figures 1 is way too abstract given the complicated set-up of the proposed architecture. It would be nice to see more details on the subnet for depth estimator and output of the net.
Overall it would be helpful for reproducibility if authors can visualize all the layers of all the different parts of the network as it is commonly done in the DL papers.
-	Talking about proposed formulation of BA use either of the following and be consistent across the paper:
Featuremetric BA / Feature-metric BA / Featuremetric BA / ‘Feature-metric BA’
-	Talking about depth parametrization use ‘basis’ or ‘bases’ not both and clearly defined the meaning of this important notion.
-	Attention should be given to the notation in formulas (3) and (4). The projection function there is no longer accepts a 3D point parametrized by 3 variables. Instead only depth is provided.
In addition, the subindex ‘1’ of the point ‘q’ is not explained.
-	More attention should be given to the evaluation section. Specifically to the tables (1 and 2) with quantitative results showing the comparison to other methods.
It is not clear how the depth error is measured and it would be nicer to have the other errors explained exactly as they referred in the tables (e.g. ATE?).
-	How the first camera pose is initialized?
-	In Figure 2.b I’m surprised by the difference obtained in the feature maps for images which seems very similar (only the lighting seems to be different). Is it three consecutive frames?
-	Attention should be given to the grammar, formatting in particular the bibliography.

---

> ### Author Response · Authors · 2018-11-27
> **Response to Reviewer 2**
>
>
> We thank the reviewer for the comments. We have revised the paper according to the suggestions and would like to clarify several things:
>
> Q1. Evaluation Time:
> We have added the detailed running time for each component in Table 3 in Appendix A of the revised version.
>
> Q2. Implementation Details:
> We will share all the source code to make sure it is reproducible. Meanwhile, we have included more details as suggested in Appendix A, including a visualization of all layers of the different parts of the network. If 1-2 extra pages are allowed, we can include those details to the paper.
>
> Q3. Figure 1 is too abstract:
> We have updated the figure to make it more intuitive and contains more details.
>
> Q4. The top row of Figure 2b is confusing:
> We apologize for the confusion caused. Shown at the top row of Figure 2b are not three consecutive frames. They are the R, G, B channels of a single frame. To avoid confusing, we use different colors for them and explained that in the figure.
>
> Q5. How the first camera pose is initialized?:
> All the camera pose including the first camera are initialized with identity rotation and zero translation, which are aligned with the coordinate system of the first camera. We clarified this at the end of Section 4.3 in the revised version.
>
> Q6. Evaluation metrics are not clear:
> To facilitate comparisons with other methods, we use the evaluation metrics in previous works in Table 1 and 2, so that we can cite the results of previous methods. As we described in the paper, the depth metric are the same as Eigen and Fergus (2015). The translation metrics(ATE) are the same as [Wang et al. 2018, Zhou et al. 2017]. In the revised version, we briefly introduce the definition of these metrics at the beginning of each paragraph in Section 5.2.
>
> Q7. Attention should be given to the notation in formulas (3) and (4):
> We changed the parameters from ‘d’ to ‘d \cdot p’ which is a 3D point. We also removed the redundant subindex ‘1’, because all points ‘q’ are on the first frame.
>
> Q8. Terminology consistency through the paper:
> Thanks for the suggestion. We consistently use the term “feature-metric BA” and “basis depth maps” through the paper now.
>
> Q9. Typos, Grammar, Format, and Bibliography:
> Thanks for pointing them out. We have revised the paper to fix these problems.

---

> > ### Comment · AnonReviewer3 · 2018-12-03
> > **The response has addressed enough of my concerns**
> >
> > The response has addressed enough of my concerns  and I determine to increase my rating from 6 to 7.

---

### Official Review · AnonReviewer1 · 2018-11-02
**Very well written paper on an important subject, with clear technical contribution and convincing results**

**Rating:** 9
**Confidence:** 4

**Review:**

This paper presents a novel approach to bundle adjustment, where traditional geometric optimization is paired with deep learning.
Specifically, a CNN computes both a multi-scale feature pyramid and a depth prediction, expressed as a linear combination of "depth bases".
These values are used to define a dense re-projection error over the images, akin to that of dense or semi-dense methods.
Then, this error is optimized with respect to the camera parameters and depth linear combination coefficients using Levenberg-Marquardt (LM).
By unrolling 5 iterations of LM and expressing the dampening parameter lambda as the output of a MLP, the optimization process is made differentiable, allowing back-propagation and thus learning of the networks' parameters.

The paper is clear, well organized, well written and easy to follow.
Even if the idea of joining BA / SfM and deep learning is not new, the authors propose an interesting novel formulation.
In particular, being able to train the CNN with a supervision signal coming directly from the same geometric optimization process that will be used at test time allows it to produce features that  will make the optimization smoother and the convergence easier.
The experiments are quite convincing and seem to clearly support the efficacy of the proposed method.

I don't really have any major criticism, but I would like to hear the authors' opinions on the following two points:

1) In page 5, the authors write "learns to predict a better damping factor lambda, which gaurantees that the optimziation will converged to a better solution within limited iterations".
I don't really understand how learning lambda would _guarantee_ that the optimization will converge to a better solution.
The word "guarantee" usually implies that the effect can be somehow mathematically proved, which is not done in the paper.

2) As far as I can understand, once the networks are learned, possibly on pairs of images due to GPU memory limitations, the proposed approach can be easily applied to sets of images of any size, as the features and depth predictions can be pre-computed and stored in main system memory.
Given this, I wonder why all experiments are conducted on sets of two to five images, even for Kitti where standard evaluation protocols would demand predicting entire sequences.

---

> ### Author Response · Authors · 2018-11-27
> **Response to Reviewer 1**
>
>
> We thank the reviewer for the comments and appreciation, and would like to answer the reviewer’s questions as follows:
>
> Q1. The use of the word “guarantees” is imprecise:
> Thanks for pointing out this. We have adjusted the word. A theoretical analysis will be an interesting future work.
>
> Q2. Whole sequence reconstruction results:
> Our current implementation only allows up to 5 images in a single 2015 TITANX GPU with 12GB memories. This is because we implemented the whole pipeline using tensorflow in python, which is memory inefficient, especially during training. Each image takes about 2.3GB memory on average, and most of the memory is consumed by the CNN features and matrix operation. But it is straightforward to concatenate multiple 5-frame segments to reconstruct a complete sequence, which is demonstrated in the comparison with CodeSLAM in Figure 7 of the revised version.  It is also straightforward to implement our BA-Layer in CUDA directly to reduce the memory consumption of matrix operation and push the number of frames.

---

### Author Response · Authors · 2018-11-27
**Response to all reviewers**

We thank all the reviewers for their insightful comments. We have revised the paper as suggested by the reviewers, and summarize the major changes as follows:

* Network architecture details and evaluation time required by Reviewer2 are added as Appendix A.

* The Figure 1. is updated  to include more details as required by Reviewer2.

*Ablation studies comparisons with Gauss-Newton and different constant lambda value required by Reviewer3  are updated in Appendix B.

*Comparison with CodeSLAM on EuroC required by Reviewer3 are updated in Appendix E.

We also would like to ask for the reviewers’ suggestions if it is allowed to have one more extra page to include more details and comparisons, and make the paper more informative to ensure reproducibility.  We targeted at 8 pages in the initial submission, but according to the reviewers’ comments, it will be helpful to have more details in the main text

The other concerns raised by the reviewers have also been addressed individually.

---

### Meta-Review · Area_Chair1 · 2018-12-11
**Nice work combining Bundle Adjustment and Deep Learning Methods**

**Confidence:** 4
**Recommendation:** Accept (Oral)

**Metareview:**

The first reviewer summarizes the contribution well: This paper combines [a CNN that computes both a multi-scale feature pyramid and a depth prediction, which is expressed as a linear combination of "depth bases"]. This is used to [define a dense re-projection error over the images, akin to that of dense or semi-dense methods]. [Then, this error is optimized with respect to the camera parameters and depth linear combination coefficients using Levenberg-Marquardt (LM). By unrolling 5 iterations of LM and expressing the dampening parameter lambda as the output of a MLP, the optimization process is made differentiable, allowing back-propagation and thus learning of the networks' parameters.]

Strengths:
While combining deep learning methods with bundle adjustment is not new, reviewers generally agree that the particular way in which that is achieved in this paper is novel and interesting. The authors accounted for reviewer feedback during the review cycle and improved the manuscript leading to an increased rating.

Weaknesses:
Weaknesses were addressed during the rebuttal including better evaluation of their predicted lambda and comparison with CodeSLAM.

Contention:
This paper was not particularly contentious, there was a score upgrade due to the efforts of the authors during the rebuttal period.

Consensus:
This paper addresses an interesting  area of research at the intersection of geometric computer vision and deep learning and should be of considerable interest to many within the ICLR community. The discussion of the paper highlighted some important nuances of terminology regarding the characterization of different methods. This paper was also rated the highest in my batch. As such, I recommend this paper for an oral presentation.